# Prevalence and associated factors of acute respiratory infection among street sweepers and door-to-door waste collectors in Dessie City, Ethiopia: A comparative cross-sectional study

Betelhiem Eneyew[1], Tadesse Sisay[1], Adinew Gizeyatu[1], Mistir Lingerew[1], Awoke Keleb[1], Asmamaw Malede[1], Ayechew Ademas[1], Mengesha Dagne[1], Mesfin Gebrehiwot[1], Yitayish Damtie[2], Tesfaye Birhane Tegegne[2], Elsabeth Addisu[2], Zinabu Fentaw[3], Birhanu Wagaye[4], Alelgne Feleke[1], Seada Hassen[1], Gete Berihun[1], Masresha Abebe[1], Leykun Berhanu[1], Tarikuwa Natnael[1], Mohammed Yenuss[1], Gebremariam Ketema[5], Kassahun Bogale[5], Tilaye Matebe Yayeh[6], Maru Selamsew[7], Alemwork Baye[8], Metadel Adane[1]*

1 Department of Environmental Health, College of Medicine and Health Science, Wollo University, Dessie, Ethiopia, 2 Department of Reproductive and Family Health, School of Public Health, College of Medicine and Health Sciences, Wollo University, Dessie, Ethiopia, 3 Department of Epidemiology and Biostatistics, School of Public Health, College of Medicine and Health Sciences, Wollo University, Dessie, Ethiopia, 4 Department of Public Health Nutrition, School of Public Health, College of Medicine and Health Sciences, Wollo University, Dessie, Ethiopia, 5 Department of Pharmacy, College of Medicine and Health Sciences, Wollo University, Dessie, Ethiopia, 6 Department of Statistics, College of Natural Sciences, Wollo University, Dessie, Ethiopia, 7 University of Gondar Comprehensive Specialized Hospital, Gondar, Ethiopia, 8 Neonatal Intensive Care Unit, Dessie Comprehensive Specialized Hospital, Dessie, Ethiopia

☯ These authors contributed equally to this work.

* metadel.adane2@gmail.com

## Abstract

### Background

Acute respiratory infections are rising in developing countries including Ethiopia. Lack of evidence for the prevalence and associated factors of acute respiratory infection among street sweepers and door-to-door waste collectors in Dessie City, Ethiopia is a challenge for the implementation of appropriate measures to control acute respiratory infection. Thus, this study was designed to address the gaps.

### Methods

A comparative cross-sectional study was conducted among 84 door-to-door waste collectors and 84 street sweepers from March to May 2018. A simple random sampling technique was used to select study participants. Data were collected by trained data collectors using a pretested structured questionnaire and on-the-spot direct observation checklist. Data were analyzed using three different binary logistic regression models at 95% confidence interval (CI): the first model (Model I) was used to identify factors associated with acute respiratory infection among street sweepers, whereas the second model (Model II) was used to identify factors associated with acute respiratory infection among door-to-door waste collectors, and the third model (Model III) was used for pooled analysis to identify factors associated with

**Data Availability Statement:** All relevant data are within the paper and its Supporting Information files.

**Funding:** Wollo University funded this study. The funders had no role in study design, data collection and analysis, decision to publish, or preparation of the manuscript.

**Competing interests:** The authors have declared that no competing interests exist.

**Abbreviations:** AOR, Adjusted Odds Ratio; ARI, Acute Respiratory Infection; CI, Confidence Interval; COR, Crude Odds Ratio; PPE, Personal Protective Equipment.

acute respiratory infection among both street sweepers and door-to-door waste collectors. From each model multivariable logistic regression, variables with a *p*-value <0.05 were taken as factors significantly associated with acute respiratory infection.

## Results

The overall prevalence of acute respiratory infection among studied population was 42.85% with 95% CI (35.1, 50.0%). The prevalence of acute respiratory infection among street sweepers was 48.80% (95% CI: 37.3, 64.8%) and among door-to-door waste collectors was 36.90% (95% CI: 27.4, 46.4%). There was no statistically significant difference between the prevalence of acute respiratory infection among the two groups due to the overlapping of the 95% CI. Among the street sweepers, we found that factors significantly associated with acute respiratory infection were not cleaning personal protective equipment after use (adjusted odds ratio [AOR]: 2.40; 95% CI: 1.15, 5.51) and use of coal/wood for cooking (AOR: 3.95; 95% CI: 1.52, 7.89), whereas among door-to-door waste collectors, were not using a nose/mouth mask while on duty (AOR: 5.57; 95% CI: 1.39, 9.32) and not receiving health and safety training (AOR: 3.82; 95% CI: 1.14–7.03) were factors significantly associated with acute respiratory infection among door-to-door-waste collectors. From the pooled analysis, we found that not using a nose/mouth mask while on duty (AOR: 2.19; 95% CI: 1.16, 4.53) and using coal/wood for cooking (AOR: 2.74; 95% CI: 1.18, 6.95) were factors significantly associated with acute respiratory infection for both street sweepers and door-to-door waste collectors.

## Conclusion

The prevalence of acute respiratory infection among street sweepers and door-to-door waste collectors has no statistically significant difference. For both groups, not using a nose/mouth mask while on duty and using coal/wood for cooking fuel factors associated with acute respiratory infection. The municipality should motivate and monitor workers use of personal protective equipment including masks and gloves. Workers should use a nose/mouth mask while on duty and should choose a clean energy source for cooking at home.

## Background

Acute respiratory infections (ARIs), which are caused by viruses and bacteria that affect the upper and lower respiratory tract, are a leading cause of morbidity and mortality globally, accounting for approximately 5.8 million deaths worldwide in 2010 [1]. Acute respiratory tract illnesses are the most frequent illnesses in humans and are an important cause of disability and days lost from school or work [2, 3].

Industrialized countries have significantly reduced occupational health impacts on street sweepers and waste collectors by applying standardized waste management processes including regular use of closed containers for waste collection [4]. In developing countries, waste that is put out for collection is rarely stored in closed containers. Rather, it is dumped in the open or left in an open carton or basket, requiring it to be picked up by hand. Therefore, workers in developing countries will have more direct contact with solid waste than their counterparts in developed countries [5].

Bioaerosols liberated from waste and compost may contain bacteria, spore forms of bacteria and fungi [6, 7]. Exposure to bioaerosols is associated with health effects such as respiratory

symptoms and influenza-like symptoms [8]. Different studies have identified age, educational status, type of home cooking fuel used, prolonged duration of working hours and employment, past medical history, use of face mask as factors having an association with ARI and respiratory symptoms [9–16]. Many studies have been conducted on the prevalence of and associated factors for ARI, but the majority of the studies have been restricted to children [13, 17–22] and so ARI data related to other age groups are limited.

In this study, the primary target population was door-to-door solid waste collectors, whose health is affected by manual sorting of waste before it is disposed of at solid waste disposal sites, and street sweepers, who inhale dust swept up by their manual brooms, which aggravates respiratory problems. The prevalence of and associated factors for ARI in street sweepers and door-to-door waste collectors remain poorly understood. It is therefore vital in this study to identify the prevalence of ARI and associated factors among door-to-door waste collectors and street sweepers in Dessie City, Ethiopia.

## Materials and methods

### Study area, study design and participants

An institution-based comparative cross-sectional study was conducted during March to May 2018 in Dessie City. Dessie City is located 401 km northeast of the capital city of Addis Ababa on the high plateau in Amhara regional state, South Wollo zone.

The city sanitation, beautification and parks development department runs the solid waste management activities of the town. The solid waste management includes door-to-door waste collection and transport to the disposal site. There are 485 door-to-door waste collectors working under ten micro-enterprises and 100 street sweepers working under ten micro-enterprises recruited by the municipality to undertake this activity. Door-to-door solid waste collectors and street sweepers employed by Dessie City municipality were the source population from which the study population was systematically selected.

### Sample size and sampling procedure

The sample size was determined using double proportion population formula

$$(Z_{\alpha/2} + Z_{\beta})^2 * [P_1(1-P_1) + P_2(1-P_2)]/(P_1-P_2)^2$$

where:

- $Z_{\alpha/2}$ is the critical value of the normal distribution at α/2 (for a confidence level of 95%, α is 0.05 and the critical value is 1.96)

- $Z_{\beta}$ is the critical value of the normal distribution at β (for a power of 80%, β is 0.2 and the critical value is 0.84).

- $P_1$ is the prevalence of respiratory symptoms among street sweepers 0.689 [10].

- $P_2$ is the prevalence of respiratory symptoms among door- to-door waste collectors 0.5 considering 50% prevalence.

$$n = \frac{(1.96 + 0.84)^2 * (0.69(1 - 0.69) + 0.5(1 - 0.5))}{(0.69 - 0.5)^2} = 75.62 \sim 76$$

By considering 10% non-response rate, which is 8, then the sample size for each group was 84; finally; the overall total sample size of the study for the two groups was 168.

There were ten door-to-door waste collecting micro-and small-scale enterprises and ten street sweeping micro-and small-scale enterprises. For both street sweepers and door-to-door waste collectors, proportional allocation of sample size was employed among ten micro enterprises working on waste collection and then simple random sampling technique was used to select study participants. Finally, 168 participants were selected using simple random sampling technique (84 street sweepers and 84 door-to-door waste collectors) (Figs 1 and 2).

## Study variables

The dependent variable of this study was the presence or absence of acute respiratory infection (ARI), whereas the independent variables included socio-demographic factors, behavioral factors, housing condition, occupational and environmental factors, institutional factors, and history of past medical illness. Presence of ARI was defined by the study participant having suffered from any ARI symptoms: a cough, fever, sore throat, chest tightness, shortness of breath (wheezing or difficulty in breathing) in the preceding two weeks before the interview [14].

## Data collection

Data were collected using face-to-face interviewer-administered questionnaire and an on-the-spot observation checklist for utilization of PPE and at home housing conditions. The questionnaire was first prepared in English, translated to Amharic and then re-translated back to English to keep consistency. The questionnaire consisted of 6 parts. Part I included questions about socio-demographic data, Part II about behavioral factors, Part III about occupational and environmental factors, Part IV about institutional factors, part V about past medical illness and Part VI about the presence of any ARI symptoms in the preceding two weeks.

Four data collectors and two supervisors, who had a BSc degree in environmental health were involved in the survey. They were given two days' training about data collection tools, the procedures to take written informed consent and other ethical issues. A pretest was done outside of the selected area using a number of people equal to 10% of study participants. Then the questionnaire amendment was done for its consistency based on the findings of the pretest before the actual data collection commenced. The supervisors regularly monitored data collectors. The completeness and consistency of the questionnaires was checked daily during data collection.

## Data analysis

Data were entered into EpiData version 3.1 and exported to SPSS version 24.0 for data cleaning and analysis. During data analysis, mean and standard deviation (SD) (mean ±SD) were calculated for continuous variables, whereas descriptive statistics of frequencies (*n*) and percentage (%) were conducted for categorical variables. Using the outcome variable of presence of ARI among street sweepers and door-to-door waste collectors, we estimated the prevalence of ARI separately for each groups. To examine whether there was a significant difference or no difference in the prevalence ARI among street sweepers and door-to-door-waste collectors, we estimated the 95% CI of the prevalence of ARI for the two groups by bootstrap using SPSS version 24.0. When there was overlapping of the 95% CI, we concluded that there was no significant difference of prevalence of ARI between street sweepers and door-to-door-waste collectors, whereas if there was no overlapping of the 95% CI of the prevalence of ARI among the two groups, we concluded there was a significant difference of the prevalence of ARI between them.

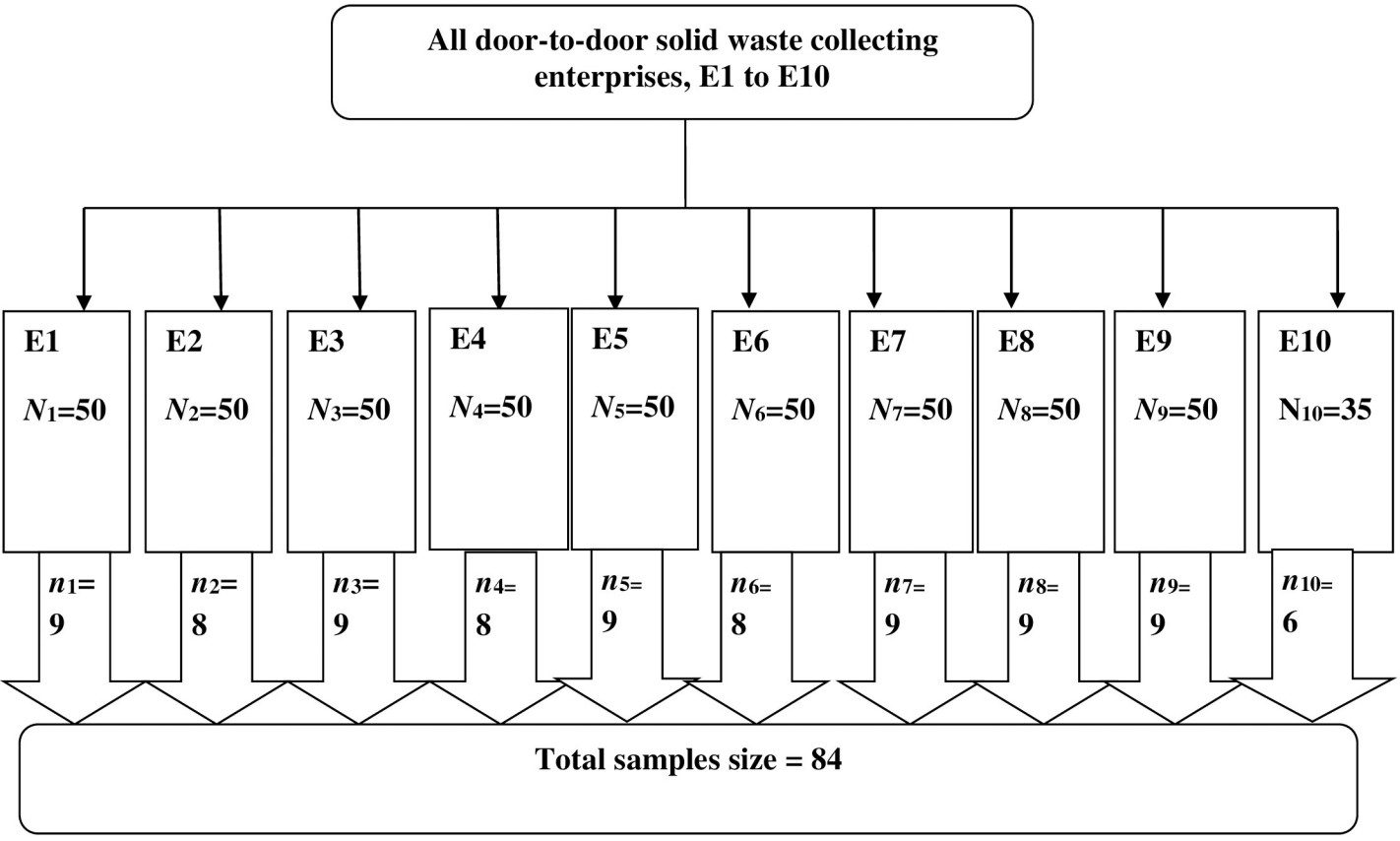

**Fig 1. Proportional allocation of sample size among enterprises of door-to-door waste collectors in Dessie City, Ethiopia, March to May 2018.**

Data were analyzed using three different binary logistic regression models at 95% CI: the first model (Model I) was used to identify factors associated with ARI among street sweepers, the second model (Model II) was used to identify factors associated with ARI among door-to-door waste collectors, and the third model (Model III) was used for pooled (combined) analysis

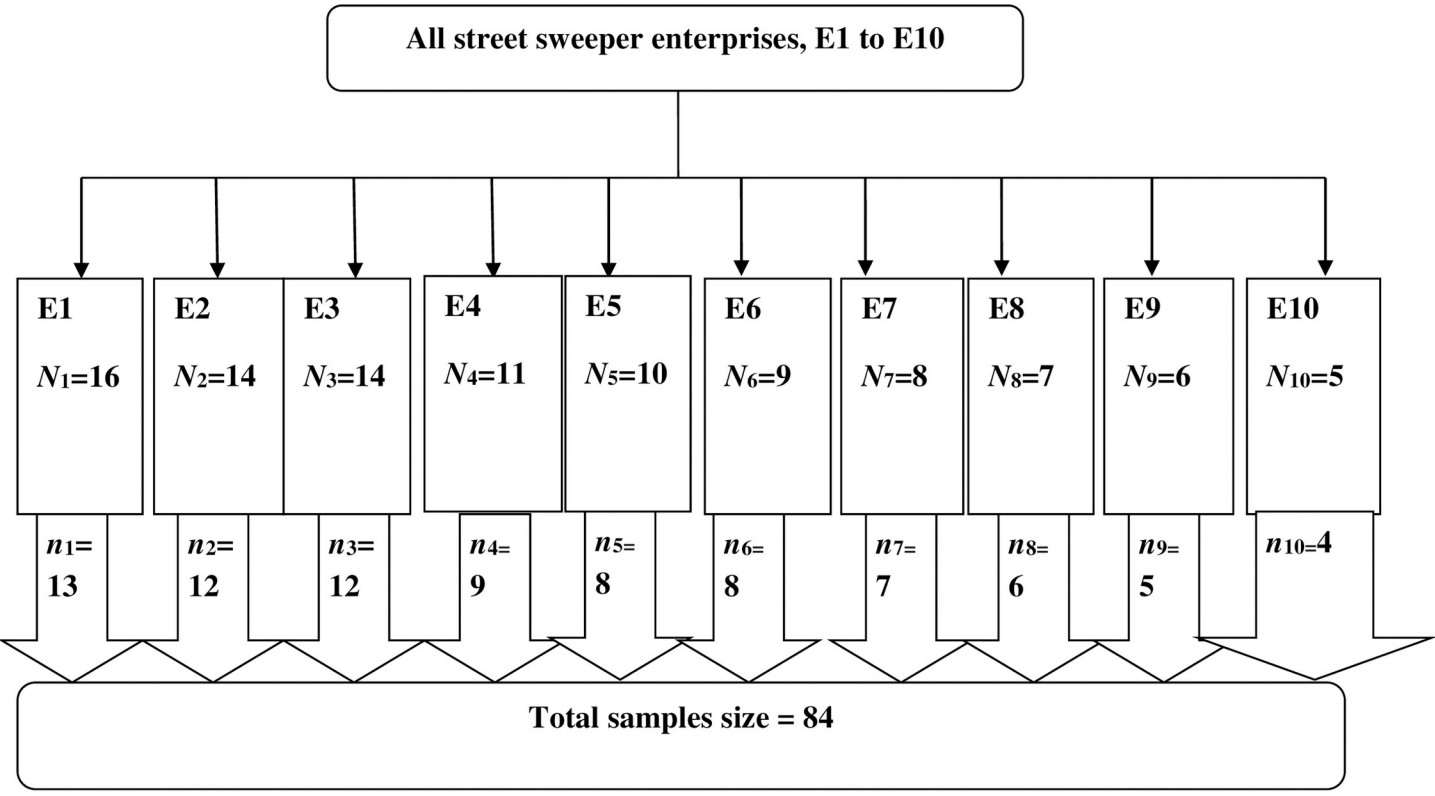

**Key:**

- E1, E2, E3, etc. = Enterprise One (EI), Enterprise Two (E2), Enterprise Three (E3) and so on to Enterprise Ten.

- $N_1$, $N_2$, $N_3$, etc. = Number of study population in each enterprise from Enterprise One to Enterprise Ten.

- $n_1$, $n_1$, $n_3$…. = Proportional allocated sample size from study population in Enterprise One to Enterprise Ten.

**Fig 2. Proportional allocation of sample size among enterprises of street sweepers in Dessie City, Ethiopia, March to May 2018.**

to identify factors associated with ARI among both street sweepers and door-to-door waste collectors. For each model, bi-variable and multivariable analysis were estimated and variables with $p < 0.25$ in bi-variable logistic regression were transferred to each adjusted model for street sweepers and door-to-door waste collectors, as well as to the pooled analysis. Finally, variables with $p$-value $< 0.05$ from Model I (street sweepers), Model II (door-to-door waste collectors) and Model III (pooled analysis) of the multivariable logistic regression were taken as factors significantly associated with ARI, respectively. Potential confounders for the three models were controlled by the adjusted analysis during multivariable logistic regression analysis.

Multi-collinearity test was carried out using standard error (SE) to see the correlation between independent variables. There was no multi-collinearity and values were 1.85, which is in the ranges of -2 < SE < 2. The Hosmer Lemeshow goodness-of-fit test [23] with *p*-value greater than 0.05 was used to check the fitness of the model; the *p*-value of each Model I, Model II and Model III was 0.931, 0.857 and 0.890, respectively.

### Ethics approval and consent to participate

Ethical clearance was obtained from Ethical Review Committee of College of Medicine and Health Sciences, Wollo University. Before conducting the survey, a supportive letter from Wollo University's Department of Environmental Health was written to the Dessie City Health Bureau and Dessie City municipality, which in turn secured permission to conduct the study. Informed written consent was obtained from the study participants after they were given information about the aim of the study. Study participants were assured that their information would not be used for purposes other than scientific research, that participation was voluntary and that they could withdraw from the interview at any time for whatever reason. Study participants who had ARI symptoms at the time of data collection were linked to the nearest health facility for treatment. Data were gathered anonymously without recording the names and any other identifiers of the study participant to keep confidentiality of the study.

## Result

### Characteristics of street sweepers and door-to-door waste collectors

Data were collected and analyzed from 84 street sweepers and 84 door-to-door waste collectors, with 100 percent response rate. The sex frequency distribution in street sweepers was male 11 (13.1%), female 73 (86.9%) and in door-to-door waste collectors was male 28 (33.3%), female 56 (66.7%). Study subjects' age range was 20–61 years. Most (91.7%) of the street sweepers and three-fourths (76.2%) of door-to-door waste collectors had a monthly income less than 27 $ USD. Among street sweepers, 22 (26.2%) had completed primary school or above, whereas 47 (56.0%) of them could not read and write (Table 1).

### Occupational and environmental factors

All study participants of both group worked 8 hours per day. Two street sweepers and 22 (26.2%) door-to-door waste collectors worked more than 48 hours per week. Among street sweepers who had worked for more than 5 years 25 (48.1%) had ARI. Of the 4 street sweepers who had past exposure to gas or chemical fumes, 2 had ARI. Only one door-to-door waste collector had past exposure to gas/chemical fumes and that participant had ARI. Of all study participants, 6 street sweepers and 8 door-to-door waste collectors had exposure to another dusty job. The number of participants with less than 5 years' work experience among street sweepers was 32 (38.0%) and among door-to-door waste collectors 77 (91.7%).

### Institutional factors

For neither category of study subjects was there a shower facility provided by the municipality. Of the study participants, masks that covered their mouths were worn by 67 (79.8%) street sweepers and 58 (69.0%) door-to-door waste collectors. The number of participants who had gloves was 67(79.8%) among street sweepers and 74 (88.1%) among door-to-door waste collectors (Fig 3). Data on the utilization of PPE while on duty at the time of data collection showed 50 (59.5%) street sweepers and 49 (58.3%) door-to-door waste collectors used a mask; and 55 (65.5%) street sweepers and 68 (81.0%) door-to-door waste collectors used gloves (Fig 4).

**Table 1. Socio-demographic characteristics among street sweepers and door-to-door waste collectors in Dessie City, Ethiopia, March to May 2018.**

| Variable | Category | ARI among street sweepers (N = 84) | | ARI door-to-door waste collectors (N = 84) | |
|---|---|---|---|---|---|
| | | Yes | No | Yes | No |
| | | *n* (%) | *n* (%) | *n* (%) | *n* (%) |
| Sex | Male | 3 (27.3) | 8 (72.7) | 11 (39.3) | 17 (60.7) |
| | Female | 38 (52.1) | 35 (47.9) | 20 (35.7) | 36 (64.3) |
| Age (years) | 18–35 | 18 (52.9) | 16 (47.1) | 22 (36.7) | 38 (63.3) |
| | 36–45 | 20 (57.1) | 15 (42.9) | 6 (31.6) | 13 (68.4) |
| | 46–61 | 3(20.0) | 12(80.0) | 3 (60.0) | 2 (40.0) |
| Marital status | Single | 7 (50.0) | 7 (50.0) | 10 (38.5) | 16 (61.5) |
| | Married | 20 (60.6) | 13 (39.4) | 11 (32.4) | 23 (67.6) |
| | Separated | 12 (42.9) | 16 (57.1) | 9 (45.0) | 11 (55.0) |
| | Widowed | 2 (22.2) | 7 (77.8) | 1 (25.0) | 3 (75.0) |
| Educational status | Cannot read and write | 18 (38.3) | 29 (61.7) | 18 (41.9) | 25 (58.1) |
| | Can read and write | 6 (40.0) | 9 (60.0) | 5 (41.7) | 7 (58.3) |
| | Primary school or above | 17 (77.3) | 5(22.2) | 8 (27.6) | 21 (72.4) |
| Family size (persons) | ≤2 | 4 (57.1) | 3 (42.9) | 9 (25.7) | 26 (74.3) |
| | 3–4 | 23 (55.1) | 28 (54.9) | 17 (43.6) | 22 (56.4) |
| | ≥ 5 | 14 (53.8) | 12 (46.2) | 5 (50.0) | 5 (50.0) |
| Monthly income ($ USD)* | ≤ $27 | 37 (48.1) | 40 (51.9) | 23 (35.9) | 41 (64.1) |
| | > $27 | 4 (57.1) | 3 (42.9) | 8 (40.0) | 12 (60.0) |

*$1 United States Dollars equal to 29 Ethiopian Birr (ETB) during March to May 2018.

Of all study participants who were given health and safety training, 55 (65.4%) street sweepers and 56 (66.7%) door-to-door waste collectors had been trained within the month previous to the time of data collection time; only 4 street sweepers and 4 door-to-door waste collectors had received pre-employment training. Of all study participants, 38 (45.2%) street sweepers and 58 (69.0%) door-to-door waste collectors did not clean their PPE after using them. Of those street sweepers who didn't clean PPE after use, 22 (57.9%) had ARI and of the similar category of waste collectors, 25 (43.1%) had ARI (Table 2).

## Housing conditions

There were 6 door-to-door waste collectors who were homeless. There were only 3 street sweepers and 1 door-to-door waste collector whose houses were built of cement. Study participants who had a cigarette smoker in their house included 7 street sweepers and 5 door-to-door waste collectors. Only 8 street sweepers and 3 door-to-door waste collectors had a chicken or farm animal in their house. Study participants who used coal/wood for cooking and also had ARI included 35 (54.7%) street sweepers and 24 (37.5%) door-to-door waste collectors (Table 3).

## Behavioral factors

All participant street sweepers had never smoked cigarettes while 5 door-to-door waste collectors had ever smoked cigarettes. Of those door-to-door waste collectors who smoked cigarettes, 3 had ARI. Among street sweepers, only 2 had always chewed chat and both of them had ARI. Among door-to-door waste collectors, 11 had always chewed chat, of whom 6 had ARI. Participants who had always drunk alcohol included 3 street sweepers and 7 door-to-door waste collectors.

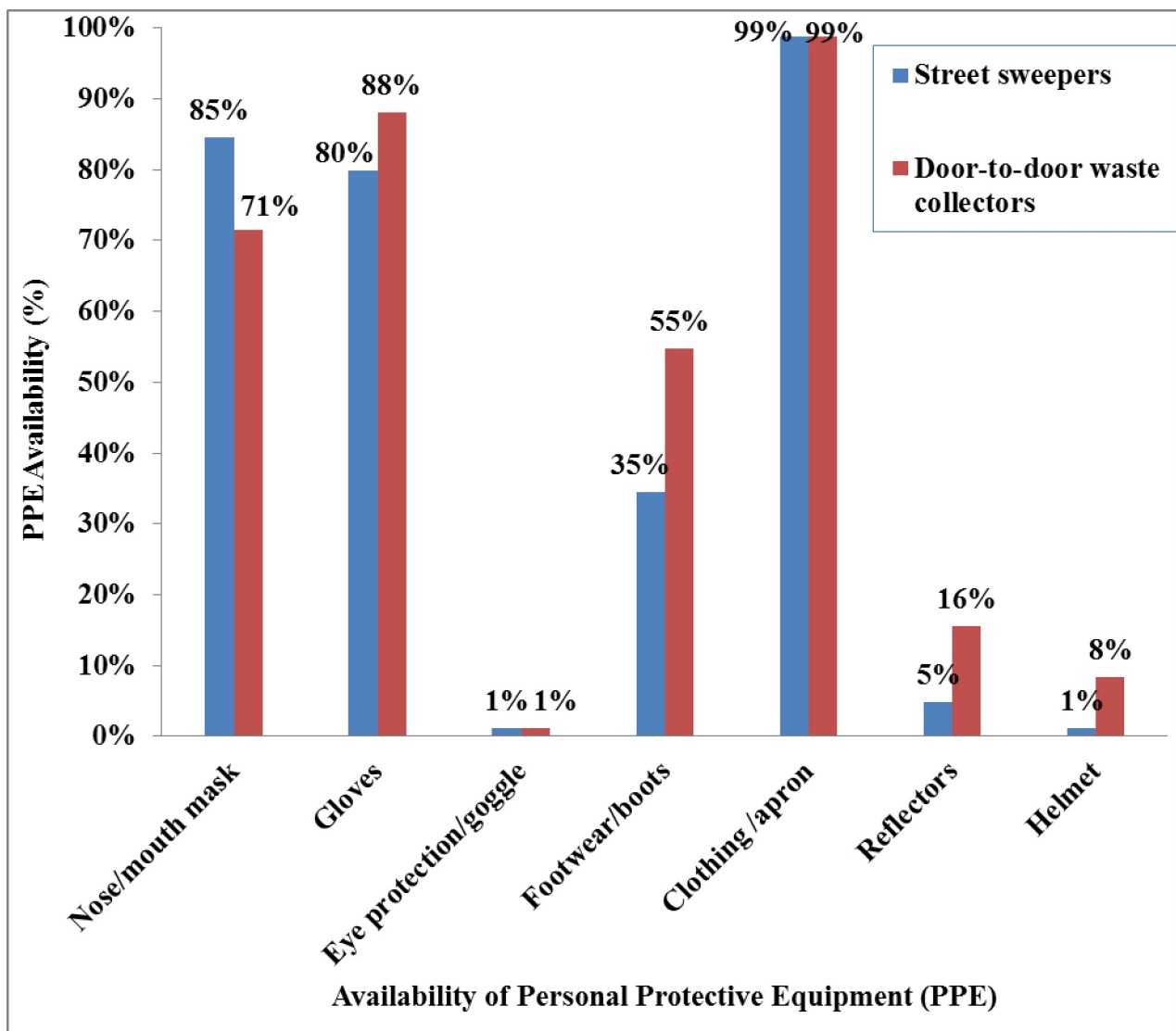

**Fig 3. Availability of personal protective equipment among street sweepers and door-to-door waste collectors in Dessie City, Ethiopia, March to May 2018.**

### Past respiratory diseases history

This study showed that, among street sweepers, reports of a history of respiratory disease included bronchitis (1 person), asthma (6 persons), and pulmonary TB (4 four persons) and 2 reports of history of heart attack; while similar reports among door-to-door waste collectors included emphysema (1 person), asthma (3 persons), pulmonary TB (2 persons), pneumonia (4 persons), sinus problems (1 person) and 3 reports of history of heart attack (Table 4).

### Prevalence and symptoms of ARI among street sweepers and door-to-door waste collectors

The prevalence of ARI among street sweepers was 48.8% with 95% CI (37.3, 64.8%) and among door-to-door waste collectors 36.9% with 95% CI (27.4, 46.4%). The overall prevalence of ARI among the studied population was 42.85% with 95% CI (35.1, 50.0%). The 95% CI

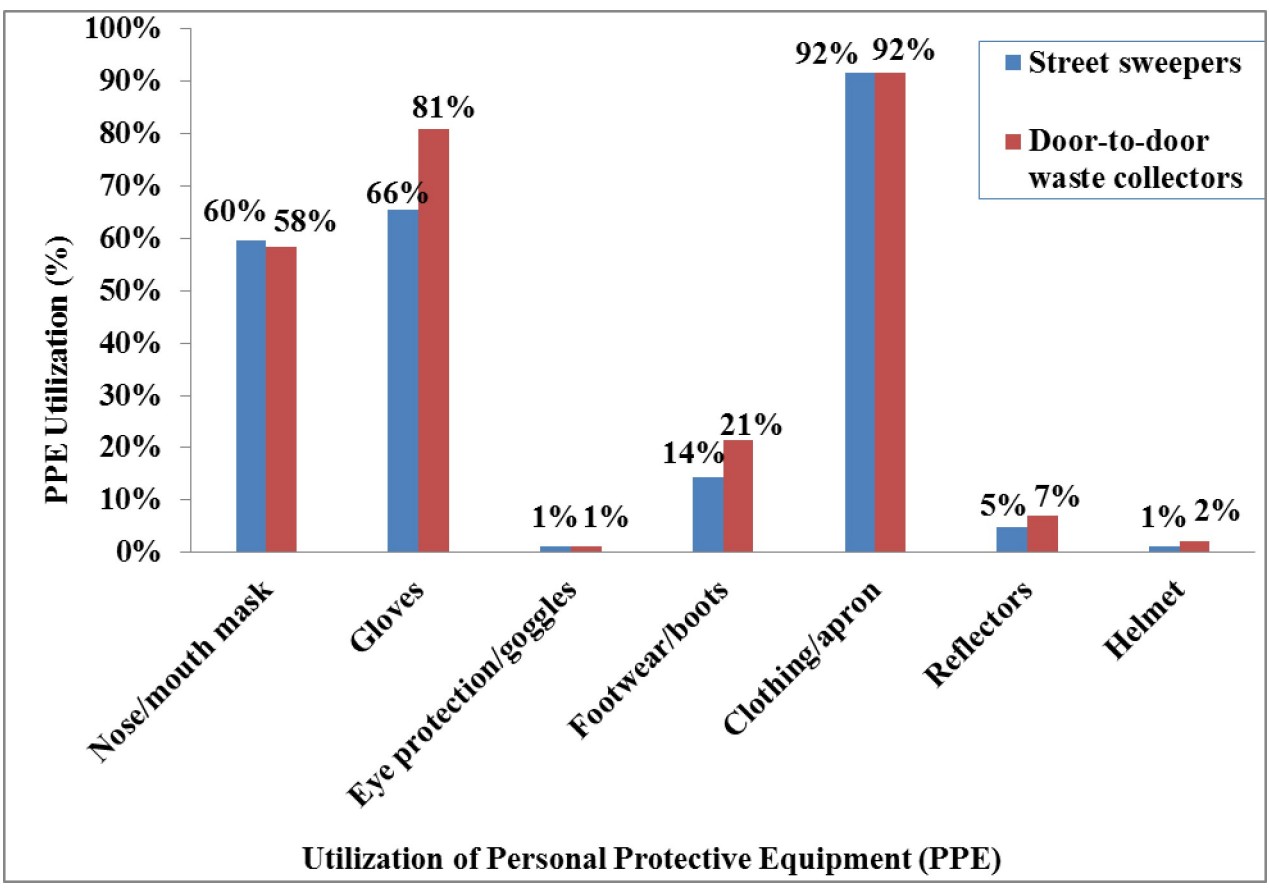

**Fig 4. Utilization of personal protective equipment among street sweepers and door-to-door waste collectors in Dessie City, Ethiopia, March to May 2018.**

prevalence of ARI among street sweepers and door-to-door waste collectors was overlapping, which indicated that there was no statistically significant difference in ARI between the two groups (Fig 5). The presence of fever was found in 28 (33.3%) among street sweepers and 16 (19.0%) door-door waste collectors. The prevalence of chest tightness among street sweepers

**Table 2. Cleaning personal protective equipment (PPE), taking shower after work and training among street sweepers and door-to-door waste collectors in Dessie City, Ethiopia March to May 2018.**

| Variable | Category | ARI among street sweepers (N = 84) | | ARI among door-to-door waste collectors (N = 84) | |
|---|---|---|---|---|---|
| | | **Yes** | **No** | **Yes** | **No** |
| | | ***n* (%)** | ***n* (%)** | ***n* (%)** | ***n* (%)** |
| Takes shower after work | No | 22 (51.2) | 21 (48.8) | 21 (35.7) | 25 (54.3) |
| | Yes | 19 (36.3) | 22 (53.7) | 10 (36.3) | 28 (73.7) |
| Cleans PPE after using it | No | 22 (57.9) | 16 (42.1) | 25 (43.1) | 33 (56.9) |
| | Yes | 19 (41.3) | 27 (58.7) | 6 (23.1) | 20 (76.9) |
| Health and safety training received | No | 12 (41.4) | 17 (58.6) | 12 (60.0) | 8 (40.0) |
| | Yes | 29 (52.7) | 26 (47.3) | 19 (29.7) | 45 (70.3) |
| Who gave training | Health professionals | 19 (48.9) | 21 (52.5) | 13 (24.5) | 40 (75.5) |
| | Our boss | 10 (66.7) | 5 (33.3) | 6 (54.5) | 5 (45.5) |

**Table 3. Housing conditions among street sweepers and door-to-door waste collectors in Dessie City, Ethiopia March to May 2018.**

| Variable | ARI among street sweepers (N = 84) | | ARI among door-to-door waste collectors (N = 84) | |
|---|---|---|---|---|
| | Yes | No | Yes | No |
| | *n* (%) | *n* (%) | *n* (%) | *n* (%) |
| Fuel used at home | 6 (37.5) | 14(62.5) | 4 (28.6) | 10 (71.4) |
| | 35 (54.7) | 29 (45.3) | 24 (37.5) | 40 (62.5) |
| Floor material | 5 (50.0) | 5 (50.0) | 3 (42.8) | 4 (57.1) |
| | 36 (48.6) | 38 (51.4) | 25 (35.1) | 46 (64.8) |
| Kitchen location | 34 (48.6) | 36 (51.4) | 25 (37.3) | 42 (62.7) |
| | 7 (50.0) | 7 (50.0) | 3 (27.3) | 8 (72.7) |
| Bedroom window | 21 (53.8) | 18 (46.2) | 17 (37.8) | 28 (62.2) |
| | 20 (54.4) | 25 (55.6) | 11 (33.7) | 22 (66.7) |
| Pets (dog or cat) in house | 31 (48.4) | 33 (51.6) | 7 (37.7) | 43 (62.3) |
| | 10 (50.0) | 10 (50.0) | 2 (23.2) | 26 (77.8) |
| Persons per room | 11 (52.4) | 10 (47.6) | 9 (25.7) | 26 (74.3) |
| | 30 (47.6) | 33 (52.4) | 19 (44.2) | 24 (55.8) |

was 15 (17.9%), twice as high as among door-to-door waste collectors 7 (8.3%). The prevalence of cough among street sweepers was 18 (21.4%) and among door-to-door waste collectors 16 (19.0%) (Table 5).

## Factors associated with ARI among street sweepers

From the multivariable analysis of data about street sweepers only, we found that street sweepers who had not cleaned PPE after use were more likely to develop ARI (AOR: 2.40; 95% CI: 1.15, 5.51) than those who cleaned PPE after use. Furthermore, we also found that those street sweepers who used coal/wood for cooking were more likely to develop ARI (AOR: 3.95; 95% CI: 1.52, 7.89) than who those who used electricity for cooking (Table 6).

## Factors associated with ARI among door-to-door waste collectors

From the multivariable analysis of using data about door-to-door waste collectors only, we found that door-to-door waste collectors who did not use a nose/mouth mask while on duty were more likely to have ARI (AOR: 5.57; 95% CI: 1.39, 9.32) than those who used a nose/mouth mask. Door-to-door waste collectors, who had not received training about health and

**Table 4. History of past respiratory disease among street sweepers and door-to-door waste collectors in Dessie City, March to May 2018.**

| Past history of respiratory disease | Street sweepers (N = 84) | Door-to-door waste collectors (N = 84) |
|---|---|---|
| | *N* | *n* |
| Bronchitis | 1 | 0 |
| Asthma | 6 | 3 |
| Pulmonary TB | 4 | 2 |
| Heart attack | 4 | 3 |
| Emphysema | 0 | 1 |
| Pneumonia | 0 | 4 |
| Sinus problems | 0 | 1 |

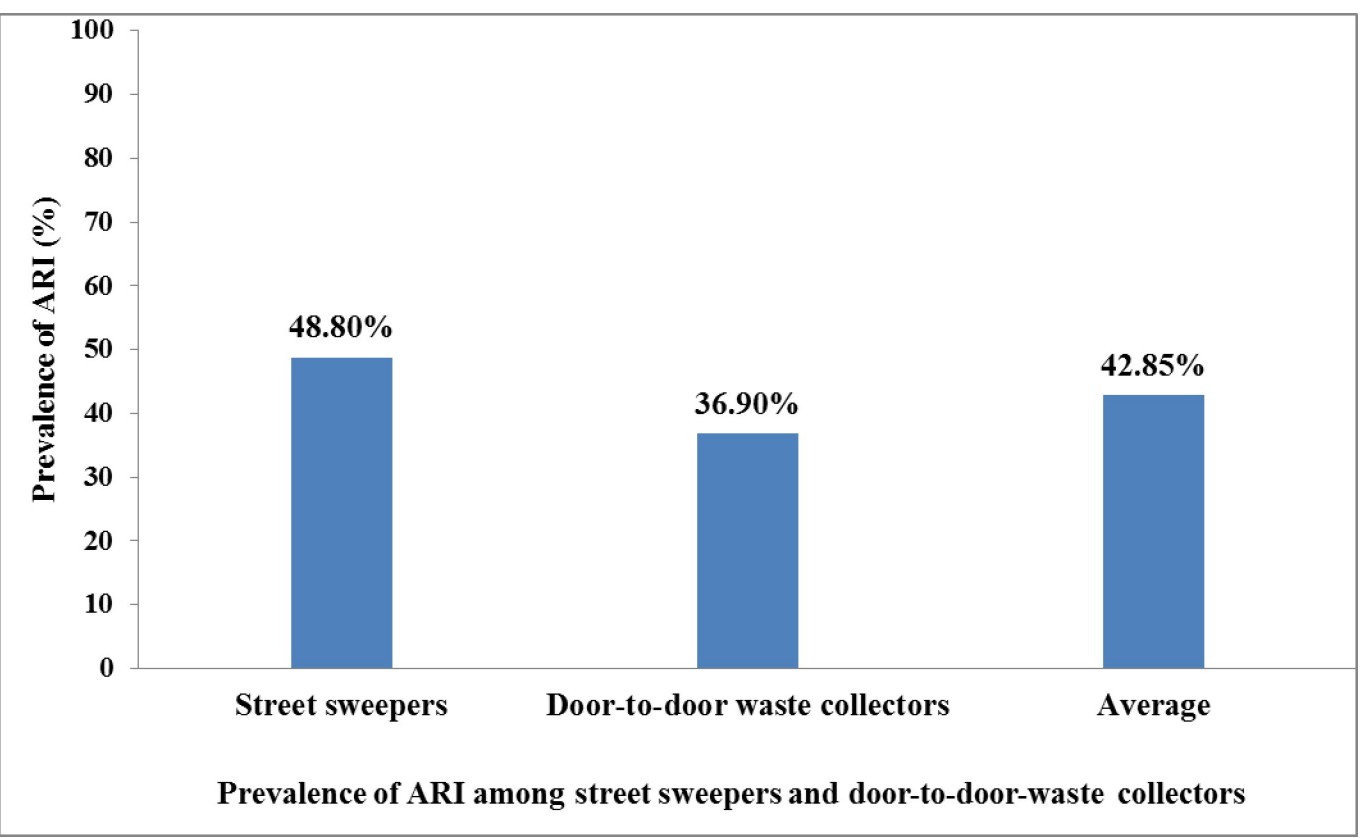

**Fig 5. Prevalence of acute respiratory infection among street sweepers and door-to-door waste collectors in Dessie City, Ethiopia, March to May 2018.**

safety were more likely to develop ARI (AOR: 3.82; 95% CI: 1.14–7.03) than those who had received training about health and safety (Table 7).

## Factors associated with ARI among both street sweepers and door-to door waste collectors from pooled multivariable analysis

From the pooled analysis of both street sweepers and door-to-door waste collectors, we found that not using a nose/mouth mask while on duty and using coal/wood for home cooking were statistically associated with ARI for both street sweepers and door-to-door waste collectors.

**Table 5. Symptoms of ARI among street sweepers and door-to-door waste collectors in Dessie city, Ethiopia, March to May 2018.**

| ARI symptoms* | Street sweepers ($N = 84$) | Door-to-door waste collectors ($N = 84$) |
|---|---|---|
| | n (%) | n (%) |
| Runny nose | 26 (31.0) | 20 (23.8) |
| Cough | 18 (21.4) | 16 (19.0) |
| Fever | 28 (33.3) | 16 (19.0) |
| Sore throat | 21 (25.0) | 20 (23.8) |
| Chest tightness | 15 (17.9) | 7 (8.3) |
| Wheezing | 16 (19) | 4 (4.8) |

*Study participants may have more than one symptom.

**Table 6. Factors associated with ARI among street sweepers from multivariable logistic regression analysis in Dessie City, Ethiopia, March to May 2018.**

| Variable | Category | Frequency | | | Model I: Street sweepers | |
|---|---|---|---|---|---|---|
| | | | ARI | | | |
| | | | Yes | No | COR (95% CI) | AOR (95% CI) |
| | | *n* (%) | *n* | *n* | | |
| Sex | Female | 73 (86.9) | 38 | 35 | 2.90 (0.71, 11.79) | 7.68 (0.48, 23.46) |
| | Male | 11 (13.1) | 3 | 8 | 1 | 1 |
| Age (years) | 18–35 | 34 (40.5) | 18 | 16 | 1 | 1 |
| | 36–45 | 35 (41.7) | 20 | 15 | 1.19 (0.46, 3.06) | 3.14 (0.60, 16.40) |
| | 46–61 | 15 (17.8) | 3 | 12 | 0.22 (0.05, 0.93) | 0.11 (0.01, 2.01) |
| Educational status | Cannot read and write | 47 (56.0) | 18 | 29 | 0.18 (0.06, 0.58) | 0.16 (0.02, 1.09) |
| | Can read and write | 15 (17.8) | 6 | 9 | 0.20 (0.05, 0.82) | 0.64 (0.051, 7.57) |
| | Primary school or above | 22 (26.2) | 17 | 5 | 1 | 1 |
| Takes shower after work | No | 43 (51.2) | 22 | 21 | 1.21 (0.52, 2.86) | 0.18 (0.02, 1.66) |
| | Yes | 41 (48.8) | 19 | 22 | 1 | 1 |
| Cleaning PPE after use | No | 38 (45.2) | 22 | 16 | 1.95 (0.98, 4.67) | 2.40 (1.15, 5.51) |
| | Yes | 46 (54.8) | 19 | 27 | 1 | 1 |
| Home used energy | Coal(wood) | 64 (76.2) | 35 | 29 | 2.92 (1.96, 8.26) | 3.95 (1.52, 7.98) |
| | Electricity | 16 (23.8) | 6 | 10 | 1 | 1 |

1, reference category.

Street sweepers and door-to-door waste collectors who did not use a nose/mouth mask while on duty were more likely to have ARI (AOR: 2.19; 95% CI: 1.16, 4.53) than those who used a nose/mouth mask. Those street sweepers and door-to-door waste collectors who used coal/wood for cooking were more likely to have ARI (AOR: 2.74; 95% CI: 1.18, 6.95) than those who used electricity for cooking (Table 8).

**Table 7. Factors associated with ARI among door-to-door waste collectors from multivariable logistic regression analysis in Dessie City, Ethiopia, March to May 2018.**

| Variable | Category | Frequency | | | Model I1: Door -to-door waste collectors | |
|---|---|---|---|---|---|---|
| | | | ARI | | COR (95% CI) | AOR (95% CI) |
| | | | Yes | No | | |
| | | *n* (%) | *n* | *n* | | |
| Family size (persons) | ≤ 2 | 35 (41.7) | 9 | 26 | 1 | 1 |
| | 3–4 | 39 (46.4) | 17 | 22 | 2.23 (0.83, 5.99) | 2.08 (0.67, 6.46) |
| | ≥ 5 | 10 (11.9) | 5 | 5 | 2.89 (0.68, 12.35) | 4.50 (0.84, 24.07) |
| Has nose/mouth masks | No | 26 (31.0) | 12 | 14 | 1.76 (0.68, 4.53) | 0.32 (0.07, 1.43) |
| | Yes | 58 (69.0) | 19 | 39 | 1 | 1 |
| Uses nose/mouth mask | No | 35 (41.7) | 20 | 15 | 4.61 (1.79, 11.89) | 5.57 (1.39, 9.32) |
| | Yes | 49 (58.3) | 11 | 38 | 1 | 1 |
| Takes shower after work | No | 46 (54.8) | 21 | 25 | 2.35 (0.93, 5.94) | 1.67 (0.57, 4.92) |
| | Yes | 38 (45.2) | 10 | 28 | 1 | 1 |
| Cleans PPE after use | No | 58 (69.0) | 25 | 33 | 2.53 (0.88, 7.22) | 1.23 (0.36, 4.24) |
| | Yes | 26 (31.0) | 6 | 20 | 1 | 1 |
| Received health and safety training | No | 20 (23.8) | 12 | 8 | 3.55 (1.25, 10.08) | 3.82 (1.14, 7.03) |
| | Yes | 64 (76.2) | 19 | 45 | 1 | 1 |

1, reference category.

**Table 8. Factors associated with ARI among both street sweepers and door-to-door waste collectors from pooled analysis of the multivariable logistic regression analysis in Dessie City, Ethiopia, March to May 2018.**

| Variable | Category | Model III: Pooled analysis (combined) | | | |
|---|---|---|---|---|---|
| | | ARI | | COR (95% CI) | AOR (95% CI) |
| | | Yes | No | | |
| | | *n* | *n* | | |
| Occupation | Street sweepers | 41 (48.8) | 43 (51.2) | 1.63 (0.88, 3.02) | 1.44 (0.59, 3.50) |
| | Door-to-door waste collectors | 31 (36.9) | 53 (63.1) | 1 | 1 |
| Family size (persons) | ≤ 2 | 13 (31.0) | 29 (69.0) | 1 | 1 |
| | 3–4 | 40 (44.4) | 50 (55.6) | 1.79 (0.82, 3.88) | 2.48 (0.66, 9.33) |
| | ≥ 5 | 19 (52.8) | 17 (47.2) | 2.49 (0.99, 6.29) | 3.74 (0.84, 16.70) |
| Working hour per week | > 48 | 7 (29.2) | 17 (70.8) | 0.50 (0.20, 1.28) | 0.46 (0.15, 1.42) |
| | ≤ 48 | 65 (45.1) | 79 (54.9) | 1 | 1 |
| Duration of occupation (years) | ≤ 5 | 43 (39.4) | 66 (60.6) | 1 | 1 |
| | > 5 | 29 (49.2) | 30 (50.8) | 1.48 (0.78, 2.81) | 1.10 (0.47, 2.55) |
| Uses nose/mouth mask | No | 38 (55.1) | 31 (44.9) | 2.34 (1.25, 4.40) | 2.19 (1.16, 4.53) |
| | Yes | 34 (34.3) | 65 (65.9) | 1 | 1 |
| Takes shower after work | No | 43 (48.9) | 45 (51.1) | 1.68 (0.91, 3.12) | 1.13 (0.52, 2.47) |
| | Yes | 29 (36.3) | 51 (63.8) | 1 | 1 |
| Cleans PPE after use | No | 47 (49.0) | 49 (51.0) | 1.80 (0.96, 3.38) | 1.81 (0.82, 3.99) |
| | Yes | 25 (34.7) | 47 (65.3) | 1 | 1 |
| Chews chat | Yes | 8 (61.5) | 5 (38.5) | 2.28 (0.71, 7.27) | 1.70 (0.36, 7.96) |
| | No | 64 (41.3) | 91(58.7) | 1 | 1 |
| Type of fuel used at home | Coal (wood) | 59 (46.1) | 69 (53.9) | 2.05 (1.41, 4.64) | 2.74 (1.18, 6.95) |
| | Electricity | 10 (29.4) | 24 (70.6) | 1 | 1 |
| Ratio of number of persons per room | 1–2.4 | 20 (35.7) | 36 (64.3) | 1 | 1 |
| | ≥2.5 | 49 (46.2) | 57 (53.8) | 1.55 (0.79, 3.04) | 0.67 (0.23, 2.01) |

1, reference category.

## Discussion

We conducted a comparative cross-sectional study among street sweepers and door-to-door waste collectors to examine the prevalence of ARI and associated factors. Our findings showed that the prevalence of ARI among street sweepers was 48.80% and among door-to-door waste collectors 36.90%. Our findings also indicated that there was no statistically significant difference of the prevalence of ARI between the two groups. From the pooled analysis, we found that not using a nose/mouth mask while on duty and using coal/wood for cooking were factors significantly associated with ARI for both street sweepers and door-to-door waste collectors.

The overall prevalence of ARI in this study was 42.9 percent. The prevalence of ARI is higher in children than adults [12], however the result of this study was in line with a related study conducted in school-aged children in Sohag and Qena governorates, Upper Egypt, which was 44.86% [14] and even higher than the prevalence of ARI in children under five years of age in rural and urban areas of Kancheepuram district, south India, which was 27.0% [13]. This indicates that those groups are highly vulnerable to ARI. A study conducted in Addis Ababa found the prevalence of street sweepers with respiratory symptoms was 68.9% [10], which is higher than our study. This may be due to socio-demographic factor differences and the fact that that study was on respiratory symptoms and not specifically on ARI.

The prevalence of ARI among street sweepers and among door-to-door waste collectors was not statistically different. This might be due to the fact that both study groups were

employed by the same institutions therefore their training, provision of PPE number of working days and their salary fairly similar. In addition to this, both groups were exposed to similar types of waste since people dispose of waste on the road-side. Educational status and housing conditions of both groups were similar. Furthermore, a lack of statistically significant differences in prevalence of ARI among the two groups might be due to similarity in study settings, environmental factors, the basic infrastructure of waste collectors, and similar characteristics of socio-demographic factors.

From the multivariable logistic regression analysis, we found that study participants who used coal/wood for cooking and who never used a facemask on duty were more likely to have ARI among both street sweepers and door-to-door waste collectors. This result is consistent with a previous study done in Ethiopia on respiratory symptoms among solid waste collectors, which indicated that those who did not use a facemask while on duty were more likely to have ARI [5, 24]. This might be because the nose/mouth mask prevented the entrance of pathogens and dust into the respiratory tract. Similar findings were found in a study conducted on prevalence of ARI among school-aged children in Egypt, in which children who lived in a house with the presence of a source of smoke were more likely to have ARI than children who lived in a house without any source of smoke [14].

A study conducted among young children in developing countries found, a significant increase in risk of ARI among those exposed to smoke or smokey fuels at home compared those in households that used cleaner fuels or were otherwise less exposed [20]. Other studies also found that use of biofuel for cooking is a risk factor associated with ARI [13, 15, 16, 22]. Cooking in the living room imposes a high level of indoor air pollution and suffocation that could increase the incidence ARI. Therefore, cooking in a separate kitchen appears to be important for preventing of ARI. Our findings are also supported by studies in Este Town in northwestern Ethiopia [17], Wolayta-Sodo in southern Ethiopia [25], and northeast Brazil [26] despite these studies being conducted among under-five children about pneumonia. The use of traditional cooking fuels also may increase indoor air pollution, which may increases the incidence of ARI. A study among under-five children about pneumonia in peri-urban areas of Dessie city also found that domestic fuel as the energy source for cooking was a factor associated with pneumonia [27]. Although the two studies' findings are exactly comparable, they showed a common factor with our study in Dessie.

## Limitations of the study and gaps for further research

That this study could not find the cause-and-effect relationship between the factors and ARI was considered s one of its limitations. Another limitation was that study was based on symptoms that were not based on clinical diagnosis by a physician but were reported by the study participants recall of the two weeks before the interview, which might have underestimated or overestimated the prevalence of ARI. Our study may over-report some variables due to social desirability bias during self-reporting. For instance, participants' recall bias might have influenced the result history of past medical illness. Further studies are recommended that consider avoiding these limitations through a follow-up study in order to obtain more valid findings.

Longitudinal studies covering different seasons may provide a better understanding of the occurrence of ARI among street sweepers and door-to-door waste collectors in Dessie City, which will help to design programs to prevent ARI. Further studies are essential to confirm ARI by clinical examination. A more detailed study that examines several occupation related diseases using clinical examination is urgently needed in order to identify all types of disease that may be caused as result of their solid waste collection work. Such type of study is helpful

for proper intervention purposes. We used some studies about children for discussion since there is a lack of studies among adult populations, which would provide more comparable details and a stronger discussion. Therefore, we notice that several studies should be conducted with the goal of preventing ARI among adults.

## Implication of the study to practice/policy

Urban sanitation is a concern for the government in Ethiopia in order to maintain and improve the health of urban residents and to attract visitors to urban areas. This is part of the United Nations Sustainable Development Goal 6 regarding clean water and sanitation for all, to ensure availability and sustainable management of water and sanitation for all, and Goal 11 about sustainable cities and communities, to make cities inclusive, safe, resilient and sustainable. To this end, solid waste collectors employed by urban municipality played an immense role in realizing proper urban sanitation through proper solid waste management, which in turn makes a city beautiful and attractive. As part of achieving these types of goals, the health and safety of solid waste collectors (street sweepers and door-to-door waste collectors) must be considered, including b not experiencing disease due to their work.

Availability of PPE, not using wood/coal as source of home energy, a good working environment, optimum work load, sufficient payment and healthy living/housing conditions might play an immense role in making waste collectors healthy and happy in their work. Our study will provide an input for health mangers, municipalities, and policy makers to prioritize areas to be improved in order to prevent ARI among urban municipality waste collectors. Furthermore, considering the prevalence of ARI among street sweepers and doo-to-door waste collectors, the Dessie City municipality in collaboration with governmental and non-governmental organizations, could further scale up programs that will help waste collectors themselves in trying to prevent ARI and other occupation related disease in a sustainable manner.

## Conclusions

The study revealed that the overall prevalence of ARI among street sweepers and door-to-door waste collectors was 42.85%. The identified significant factors for ARI among street sweepers and door-to-door waste collectors were not using a nose/mouth mask while on duty and using coal and wood for household cooking. We recommend that the municipality regularly provide these workers with adequate and quality PPE and motivate and monitor the workers use of PPE while they are on duty. We also recommend that street sweepers and door-to-door waste collectors use nose/mouth masks while on duty and use a clean energy source such as electricity, biogas and solar energy for cooking at home.

## Supporting information

**S1 Questionnaire. English version of the questionnaire.** Survey of prevalence and associated factors of acute respiratory infection among street sweepers and door-to-door waste collectors in Dessie City, Ethiopia.
(DOCX)

**S2 Questionnaire. Amharic (local language) version of the questionnaire.** Survey of prevalence and associated factors of acute respiratory infection among street sweepers and door-to-door waste collectors in Dessie City, Ethiopia.
(DOCX)

## Acknowledgments

We acknowledged Dessie City Health Bureau for providing information and support during the study. Our deepest gratitude goes to Dessie City Municipality Sanitation, Beautification and Park Department for their support during the study by providing the necessary information while we needed. Our heartfelt thanks also go to data collectors, supervisors and the study participant door-to-door waste collectors and street sweepers for their cooperation during data collection. Last but not the least, Lisa Penttila is also duly acknowledged for the language editing of this paper.

## Author Contributions

**Conceptualization:** Betelhiem Eneyew, Tadesse Sisay, Adinew Gizeyatu, Mistir Lingerew, Metadel Adane.

**Data curation:** Betelhiem Eneyew, Tadesse Sisay, Metadel Adane.

**Formal analysis:** Betelhiem Eneyew, Tadesse Sisay, Tilaye Matebe Yayeh, Metadel Adane.

**Funding acquisition:** Betelhiem Eneyew, Tadesse Sisay, Adinew Gizeyatu.

**Investigation:** Betelhiem Eneyew, Tadesse Sisay, Adinew Gizeyatu, Mistir Lingerew, Awoke Keleb, Asmamaw Malede, Ayechew Ademas, Mengesha Dagne, Mesfin Gebrehiwot, Yitayish Damtie, Tesfaye Birhane Tegegne, Elsabeth Addisu, Zinabu Fentaw, Birhanu Wagaye, Alelgne Feleke, Seada Hassen, Gete Berihun, Masresha Abebe, Leykun Berhanu, Tarikuwa Natnael, Mohammed Yenuss, Gebremariam Ketema, Kassahun Bogale, Tilaye Matebe Yayeh, Metadel Adane.

**Methodology:** Betelhiem Eneyew, Tadesse Sisay, Adinew Gizeyatu, Mistir Lingerew, Awoke Keleb, Asmamaw Malede, Ayechew Ademas, Metadel Adane.

**Project administration:** Betelhiem Eneyew, Tadesse Sisay, Adinew Gizeyatu, Mistir Lingerew, Awoke Keleb, Asmamaw Malede, Ayechew Ademas, Mengesha Dagne, Mesfin Gebrehiwot, Yitayish Damtie, Tesfaye Birhane Tegegne, Elsabeth Addisu, Zinabu Fentaw, Birhanu Wagaye, Alelgne Feleke, Seada Hassen, Gete Berihun, Masresha Abebe, Leykun Berhanu, Tarikuwa Natnael, Mohammed Yenuss, Gebremariam Ketema, Kassahun Bogale, Tilaye Matebe Yayeh, Maru Selamsew, Metadel Adane.

**Resources:** Betelhiem Eneyew, Tadesse Sisay, Adinew Gizeyatu, Mistir Lingerew, Awoke Keleb, Asmamaw Malede, Ayechew Ademas, Mengesha Dagne, Mesfin Gebrehiwot, Yitayish Damtie, Tesfaye Birhane Tegegne, Elsabeth Addisu, Zinabu Fentaw, Birhanu Wagaye, Alelgne Feleke, Seada Hassen, Gete Berihun, Masresha Abebe, Leykun Berhanu, Tarikuwa Natnael, Mohammed Yenuss, Gebremariam Ketema, Kassahun Bogale, Tilaye Matebe Yayeh, Maru Selamsew, Alemwork Baye, Metadel Adane.

**Software:** Betelhiem Eneyew, Tadesse Sisay, Adinew Gizeyatu, Mistir Lingerew, Awoke Keleb, Asmamaw Malede, Ayechew Ademas, Mengesha Dagne, Mesfin Gebrehiwot, Yitayish Damtie, Tesfaye Birhane Tegegne, Elsabeth Addisu, Zinabu Fentaw, Birhanu Wagaye, Alelgne Feleke, Seada Hassen, Gete Berihun, Masresha Abebe, Leykun Berhanu, Tarikuwa Natnael, Mohammed Yenuss, Gebremariam Ketema, Kassahun Bogale, Tilaye Matebe Yayeh, Maru Selamsew, Alemwork Baye, Metadel Adane.

**Supervision:** Betelhiem Eneyew, Tadesse Sisay, Adinew Gizeyatu, Mistir Lingerew, Awoke Keleb, Asmamaw Malede, Ayechew Ademas, Mengesha Dagne, Mesfin Gebrehiwot, Yitayish Damtie, Tesfaye Birhane Tegegne, Elsabeth Addisu, Zinabu Fentaw, Birhanu Wagaye,

Alelgne Feleke, Seada Hassen, Gete Berihun, Masresha Abebe, Leykun Berhanu, Tarikuwa Natnael, Mohammed Yenuss, Gebremariam Ketema, Kassahun Bogale, Tilaye Matebe Yayeh, Maru Selamsew, Alemwork Baye, Metadel Adane.

**Validation:** Betelhiem Eneyew, Tadesse Sisay, Adinew Gizeyatu, Mistir Lingerew, Awoke Keleb, Asmamaw Malede, Ayechew Ademas, Mengesha Dagne, Mesfin Gebrehiwot, Yitayish Damtie, Tesfaye Birhane Tegegne, Elsabeth Addisu, Zinabu Fentaw, Birhanu Wagaye, Alelgne Feleke, Seada Hassen, Gete Berihun, Masresha Abebe, Leykun Berhanu, Tarikuwa Natnael, Mohammed Yenuss, Gebremariam Ketema, Kassahun Bogale, Tilaye Matebe Yayeh, Maru Selamsew, Alemwork Baye, Metadel Adane.

**Visualization:** Betelhiem Eneyew, Tadesse Sisay, Adinew Gizeyatu, Mistir Lingerew, Awoke Keleb, Asmamaw Malede, Ayechew Ademas, Mengesha Dagne, Mesfin Gebrehiwot, Yitayish Damtie, Tesfaye Birhane Tegegne, Elsabeth Addisu, Zinabu Fentaw, Birhanu Wagaye, Alelgne Feleke, Seada Hassen, Gete Berihun, Masresha Abebe, Leykun Berhanu, Tarikuwa Natnael, Mohammed Yenuss, Gebremariam Ketema, Kassahun Bogale, Tilaye Matebe Yayeh, Maru Selamsew, Alemwork Baye, Metadel Adane.

**Writing – original draft:** Betelhiem Eneyew, Metadel Adane.

**Writing – review & editing:** Metadel Adane.

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
