## [Decision Letter · Decision Letter 0]

7 May 2020

PONE-D-20-05977

Proportion and Associated Factors of Acute Respiratory Infection (ARI) Among Street Sweepers and Door-to-Door Waste Collectors in Dessie City, Ethiopia: A Comparative Cross-Sectional Study

PLOS ONE

Dear Dr Adane (PhD),

Thank you for submitting your manuscript to PLOS ONE. After careful consideration, we feel that it has merit but does not fully meet PLOS ONE’s publication criteria as it currently stands. Therefore, we invite you to submit a revised version of the manuscript that addresses the points raised during the review process.

The paper looks interesting to me. However, the reviewers raise a number of issues that you should take into account for the paper to be acceptable for PLoS ONE. Apart from that, please check Table 4 as it looks to me that some information in the left column for variables is missing. I think also that a new table collecting the variables under consideration in the study and their definition is needed in the "Data collection and mesaurements" section. The definitions appearing in lines 139-142 in that section are completely disconnected with the rest of that section. In my opinion, you some further work is needed on that section.

We would appreciate receiving your revised manuscript by Jun 21 2020 11:59PM. To enhance the reproducibility of your results, we recommend that if applicable you deposit your laboratory protocols in protocols.io, where a protocol can be assigned its own identifier (DOI) such that it can be cited independently in the future. For instructions see: http://journals.plos.org/plosone/s/submission-guidelines#loc-laboratory-protocols

We look forward to receiving your revised manuscript.

Kind regards,

Miguel Alejandro Fernández, Ph.D.

Academic Editor

PLOS ONE

Journal Requirements:

2. Please provide additional details regarding participant consent. In the ethics statement in the Methods and online submission information, please ensure that you have specified what type of consent you obtained (for instance, written or verbal, and if verbal, how it was documented and witnessed). If your study included minors, state whether you obtained consent from parents or guardians. If the need for consent was waived by the ethics committee, please include this information.”

3. Please address the following:

a) Please include additional information regarding the survey or questionnaire used in the study and ensure that you have provided sufficient details that others could replicate the analyses. For instance, if you developed a questionnaire as part of this study and it is not under a copyright more restrictive than CC-BY, please include a copy, in both the original language and English, as Supporting Information. In addition, please provide further details of how this questionnaire was developed, for example by citing previous literature.

b) Please ensure you have discussed the potential impact of confounding variables

"The funders had no role in study design, data collection and analysis, decision to

publish, or preparation of the manuscript."

5. We note that Figure 1 in your submission contain map images which may be copyrighted. All PLOS content is published under the Creative Commons Attribution License (CC BY 4.0), which means that the manuscript, images, and Supporting Information files will be freely available online, and any third party is permitted to access, download, copy, distribute, and use these materials in any way, even commercially, with proper attribution. For these reasons, we cannot publish previously copyrighted maps or satellite images created using proprietary data, such as Google software (Google Maps, Street View, and Earth). For more information, see our copyright guidelines: http://journals.plos.org/plosone/s/licenses-and-copyright.

a) You may seek permission from the original copyright holder of Figure(s) [#] to publish the content specifically under the CC BY 4.0 license. 

Reviewers' comments:

Reviewer's Responses to Questions

**Comments to the Author**

1. Is the manuscript technically sound, and do the data support the conclusions?

Reviewer #1: Yes

Reviewer #2: Yes

Reviewer #3: Yes

Reviewer #4: Partly

2. Has the statistical analysis been performed appropriately and rigorously? 

Reviewer #1: Yes

Reviewer #2: Yes

Reviewer #3: Yes

Reviewer #4: Yes

3. Have the authors made all data underlying the findings in their manuscript fully available?

Reviewer #1: Yes

Reviewer #2: No

Reviewer #3: Yes

Reviewer #4: Yes

4. Is the manuscript presented in an intelligible fashion and written in standard English?

Reviewer #1: Yes

Reviewer #2: No

Reviewer #3: Yes

Reviewer #4: No

5. Review Comments to the Author

Reviewer #1: the paper is well written except some editorial mistakes. The figures included in the abstract section should be in line with the figures in the results section. For example the proportion of ARI among street sweepers (42.9% vs 48.8%). Another editorial issue is on the sample size calculation formula. In the equation za and zB should be squared.

Reviewer #2: Dear PLOS One thank you for the chance given to review a research article titled "Proportion and Associated Factors of Acute Respiratory Infection (ARI) Among Street Sweepers and Door-to-Door Waste Collectors in Dessie City, Ethiopia: A Comparative Cross-Sectional Study". acute respiratory diseases are the commonest causes of morbidity and mortality in street sweepers and in the door-to door waste collectors due to occupational exposure to different wastes. Thus, this study aimed to investigate the potential drivers of acute respiratory infection among exposed occupational risk groups. The following are my comments:

On the abstract Section:

-Sampling technique is missed

-line no 40: The overall proportion of ARI was high and no statistically significant difference was found between the two groups despite ARI being slightly higher in street sweepers. what is your standard to say is it high or low? and it is better to replace proportion with prevalence. How could you check difference b/n two groups was significant or not and it is not your scope.

- In keywords include associated factors

Introduction

The flow lacks coherence and scanty. Global, regional and national data on the matter of interest were missed?

On the Methods Section

-You did not determine sample size for the second objective?

-why you employed systematic sampling? It is unlikely for your study.

-Where did you get and/ how did you develop data collection tools your study? it is also not clear how could you measure validity and reliability of questionnaire?

-line 118-119 what is your data collection tool? is it interviewer administered questionnaire / observation checklist or both. make it explicit?

-what is your dependent variable and independent variables of the study?

Result section

-Present some of your findings eg mean/median age of participants and proportion of educational status, monthly income... and it is better to present the frequency of significant variables.

- line 167-169: The sex frequency distribution in street sweepers was male 11 (13.1%), female 73 (86.9%) and in door-to-door waste collectors was male 28 (33.3%), female 56 (66.7%). what does it mean? it lacks clarity.

- line 169: Study subjects age range was 20–61years but you did not specify the final age range in table 1 simply put as >45.

- what do that it mean when an individual is ever chewer and ever drinker ?

Discussion section

-Avoid presenting frequencies. ex: Multivariable logistic regression analysis revealed that ARI was three times more likely among individuals who did not use a facemask on the job.

- 240-41: The proportion of ARI among street sweepers and among door-to-door waste collectors was similar. what does it mean? do you have data but in the result part you presented them unlikely.

-The arguments are lack of scientific reasoning and reference?

- Implications were missed and what is the strength of your study?

On the Conclusion

-The study revealed that the overall proportion of ARI among street sweepers and door-to-door waste collectors is high. what is your standard to say high/low? you should ope-rationalize it.

-The difference in the proportion of ARI among street sweepers compared to door-to-door waste collectors was not statistically significant where do you get it? do you have data? but you did not present it in the result section and it is not your scope.

-Being a street sweeper or a door-to-door waste collector was not significantly associated with ARI. where do you got it?

- you are recommend that the municipality regularly provides these workers with adequate and quality PPEs, motivates and monitors the workers to use their PPEs while they are on duty an provides pre-employment safety training. are these significant finding affecting ARI? .

-you also recommended that these workers use a clean energy source like biogas and solar energy for cooking. what does it mean? are you considered biogas and solar energy as factors affecting ARI? In general, you should recommend according to your finding and practicality or applicability of result not from ground science.

-you did not incorporate declarations in your manuscript as well.

Reviewer #3: You raised an important issue which is a common problem of developing countries including Ethiopia. Overall, the document is well written but there are grammatical and writing errors which has to be corrected in addition to the following specific comments and questions.

Title

• Your title is too long. Make it short. Avoid “A Comparative Cross-Sectional Study” from the title. It is not as such strong study design. It may not attract readers to go through the whole paper.

Abstract

• Remove abbreviations like ARI, CI and so on from abstract.

• Methods:

The sampling technique is not mentioned.

Instead of using the term bi-variate, use the term bi-variable. They are different concepts.

• Results:

No need of writing 95% CI for associated factors in abstract section. Mentioning which variables were significant is enough.

How the overall proportion of ARI could be the same (42.9%) with the proportion of ARI among street sweepers while there is difference in proportion of ARI among street sweepers (42.9%) and door to door waste collectors (36.9%)?

• Conclusion:

Don’t use at higher risk of ARI since your study design doesn’t allow you to measure risk.

Background:

• Use terminologies uniformly like proportion and associated factors throughout the paper. Don’t mix up with prevalence and risk factors.

Materials and Methods:

• Sample size and sampling procedure:

The assumption for allocating the sample size for each micro enterprise in both groups is not clear. Why you include all?

• Data collection and measurements:

Change risk factors by associated factors

• Ethics approval and consent to participate:

How did you maintain confidentiality? specify it.

Results:

Grammatical and write up errors should be corrected.

• Characteristics of participants:

Instead of characteristics of participants, make it characteristics of street sweepers and door-to-door waste collectors.

• Proportion of ARI and ARI Symptoms:

“The proportion of ARI among street sweepers was 48.8 percent with 95% CI (37.3, 64.8) and among door-to-door waste collectors 36.9% with 95% CI (27.4, 46.4). The overall proportion of ARI among studied population was 42.9% with 95% CI (35.1, 50.0)”. This is different from what you write in the abstract section. Correct your writing error.

• Factors Associated with ARI:

Factors for two groups are not clearly indicated. You should clearly mention which factors were significant among street sweepers and door to door waste collectors. That is expected from your study.

Discussion:

• The comparison of the proportion of ARI made is not appropriate; children are compared with adults.

Conclusions:

• You concluded that the overall proportion of ARI among street sweepers and door-to-door waste collectors is high. As you mentioned, the only comparable study done at Addis Ababa showed that the proportion of street sweepers with respiratory symptoms was 68.9%. So, how did you say that?

Table 1: Change inappropriate wording like illiterate

Table 2: Properly write the title by mentioning street sweepers and door-to-door waste collectors

Reviewer #4: 1. From the abstract do not use abrevations

2. Sample size is too small in amount,How did you conclud?

3. Your disccusion is superficial

4. Minimize Page Numbers

5.it has hramatical error,please check it

6. PLOS authors have the option to publish the peer review history of their article (what does this mean?). If published, this will include your full peer review and any attached files.

Reviewer #1: No

Reviewer #2: Yes: Erkihun Tadesse Amsalu

Reviewer #3: No

Reviewer #4: No

---

## [Author Response · Author response to Decision Letter 0]

2 Mar 2021

Date: March 02, 2021

Manuscript title: Prevalence and Associated Factors of Acute Respiratory Infection among Street Sweepers and Door-to-Door Waste Collectors in Dessie City, Ethiopia

Manuscript ID. No: PONE-D-20-05977

Corresponding author: Metadel Adane (PhD) et al. 

Dear Miguel Alejandro Fernández, Ph.D.

Academic Editor

PLOS ONE

Thank you for your letter dated May 7, 2020 with a decision of revision needed. We were pleased to know that our manuscript was considered potentially acceptable for publication in PLoS ONE, subject to adequate revision as requested by the reviewers. Based on the instructions provided in your letter, we uploaded the file of the rebuttal letter and the marked up copy of the revised manuscript highlighting the changes made in the original submitted version. 

We have revised the manuscript by modifying the abstract, introduction, methods, results, discussion and other sections, based on the comments made by the reviewers and using the journal guidelines. Accordingly, we have marked in red color all the changes made during the revision process. Appended to this letter underneath is our point-by-point response (rebuttal letter) to the comments made by the reviewers. 

We agree with almost all the comments/questions raised by the reviewers and provided justification for disagreeing with some of them. We would like to take this opportunity to express our thanks to the reviewers for their valuable comments and to thank you for allowing us to resubmit a revision of the manuscript. 

I hope that the revised manuscript is accepted for publication in PLoS ONE. 

Sincerely yours,

Metadel Adane (PhD in Water and Public Health)

Journal Requirements

Response: We thank you for your key comments and we revised the manuscript accordingly POLS ONE manuscript preparation templates including file naming (Please see the revised version).

2. Please provide additional details regarding participant consent. In the ethics statement in the Methods and online submission information, please ensure that you have specified what type of consent you obtained (for instance, written or verbal, and if verbal, how it was documented and witnessed). If your study included minors, state whether you obtained consent from parents or guardians. If the need for consent was waived by the ethics committee, please include this information.”

Response: Written consent was obtained from the study participant and please see the updated ethical statement in page 9 and 10 form lines 199 to 200. 

3. Please address the following:

a) Please include additional information regarding the survey or questionnaire used in the study and ensure that you have provided sufficient details that others could replicate the analyses. For instance, if you developed a questionnaire as part of this study and it is not under a copyright more restrictive than CC-BY, please include a copy, in both the original language and English, as Supporting Information. In addition, please provide further details of how this questionnaire was developed, for example by citing previous literature.

Response: We provided the survey tool or questionnaire as supportive information in both English and Amharic (original language) version labeled as S1 and S2, respectively (Please see on the revised version).

4. Please ensure you have discussed the potential impact of confounding variables

 Response: We controlled the confounding variables during the multivariable analysis and we elaborated this in the data analysis section in page 8 to 9 from lines 187 to 188. 

4. Financial disclosure:

Wollo University funded this study. The funders had no role in study design, data collection and analysis, decision to publish, or preparation of the manuscript.

Line by line response to reviewers

Reviewer #1:

the paper is well written except some editorial mistakes. The figures included in the abstract section should be in line with the figures in the results section. For example the proportion of ARI among street sweepers (42.9% vs 48.8%). Another editorial issue is on the sample size calculation formula. In the equation za and zB should be squared.

Response: We appreciate you scientific judgment and thank you for your positive reflection of our study. Your concerns are valid and addressed accordingly. Please see the updated version of the manuscript. 

Reviewer #2:

Dear PLOS One thank you for the chance given to review a research article titled "Proportion and Associated Factors of Acute Respiratory Infection (ARI) Among Street Sweepers and Door-to-Door Waste Collectors in Dessie City, Ethiopia: A Comparative Cross-Sectional Study". Acute respiratory diseases are the commonest causes of morbidity and mortality in street sweepers and in the door-to door waste collectors due to occupational exposure to different wastes. Thus, this study aimed to investigate the potential drivers of acute respiratory infection among exposed occupational risk groups. The following are my comments. 

Response: Many thanks for your positive reflection of our study. Your concerns are valid and addressed accordingly. 

On the abstract Section:

-Sampling technique is missed

Response: The sampling technique is included and please see page 8 in lines 132 to 133. 

-line no 40: The overall proportion of ARI was high and no statistically significant difference was found between the two groups despite ARI being slightly higher in street sweepers. what is your standard to say is it high or low? and it is better to replace proportion with prevalence. How could you check difference b/n two groups was significant or not and it is not your scope.

Response: Thank you again for these key comments. We deleted those confusing terms of saying high and low. However, replaced proportion by prevalence. To examine either there was a significant difference or no difference of the prevalence ARI among street sweepers and door-to-door-waste collectors, we estimated the 95% CI of the prevalence of ARI for the two groups using SPSS version 24.0. When there was overlapping of the 95% CI, we concluded that there was no significant difference of prevalence of ARI between street sweepers and door-to-door-waste collectors, whereas if there was no overlapping of the 95% CI of the prevalence of ARI among the two groups, there was a significant difference of the prevalence of ARI among street sweepers and door-to-door-waste collectors. 

- In keywords include associated factors

Response: It is included but in PLoS ONE during publication, keywords are not necessary. 

Introduction

The flow lacks coherence and scanty. Global, regional and national data on the matter of interest were missed?

Response: We accept the comment. However, most studies of ARI studies were common among children and our studies are among adults aged 18 years and above, which makes as unable to explore sufficient literature. Eleven some of our discussion are used studies among under five children, which is one of the limitation of this study. In any case, we tried our best to provide good introduction besides these challenges to develop a well-organized introduction (See the revised version of the introduction). 

On the Methods Section

-You did not determine sample size for the second objective?

Response: Your idea is good and reasonable. Thank you. We were already determined sample size for the second specific objective during determining the time of proposal development, but we found that sample size based on the first specific objective was higher and we took that. But it is not common to show sample size calculation each objective during publication. 

-why you employed systematic sampling? It is unlikely for your study.

Response: We did not used systematic sampling and sorry if it is written within the manuscript. We sued simple random sampling in our study (Please see the revised version of the manuscript). 

-Where did you get and/ how did you develop data collection tools your study? it is also not clear how could you measure validity and reliability of questionnaire?

-line 118-119 what is your data collection tool? is it interviewer administered questionnaire / observation checklist or both. make it explicit?

Response: We collected data using both interviewer administered questionnaire / observation checklist and we updated the manuscript (see in page 7 from lines 146 to 150. 

-what is your dependent variable and independent variables of the study?

Response: The dependent variable was the presence or absence of ARI, whereas the independent variables included socio-demographic factors, behavioral factors, housing condition, occupational and environmental factors, institutional factors, and history of past medical illness. These points were already mentioned at the study variables sub-sections within the methods during the original submission, still you can find it in the manuscript. 

Result section

-Present some of your findings e.g mean/median age of participants and proportion of educational status, monthly income... and it is better to present the frequency of significant variables.

Response: We updated the descriptive result section as suggested and please see the revised version in page 10 from lines 213 to 217. 

- line 167-169: The sex frequency distribution in street sweepers was male 11 (13.1%), female 73 (86.9%) and in door-to-door waste collectors was male 28 (33.3%), female 56 (66.7%). what does it mean? it lacks clarity.

Response: This is because of the study was a comparative study among the two groups and the sex frequency distribution was not similar. The percentage was also calculated for each group of the total 84 study participants. This is the nature of data and we could not do about the percentage of the frequency. 

- line 169: Study subjects age range was 20–61years but you did not specify the final age range in table 1 simply put as >45.

Response: Thank you for detecting such errors and we updated the manuscript by fixing the range. 

- what do that it mean when an individual is ever chewer and ever drinker ?

Response: It means that is always chewer and always drinker. We deleted ever and replaced with always. Thank you. 

Discussion section

-Avoid presenting frequencies. ex: Multivariable logistic regression analysis revealed that ARI was three times more likely among individuals who did not use a facemask on the job.

Response: Thank you, we updated as suggested and we accepted your comment, which will minimize repetitions. 

- 240-41: The proportion of ARI among street sweepers and among door-to-door waste collectors was similar. what does it mean? do you have data but in the result part you presented them unlikely.

Response: Many thanks for this pertinent comment. We mean that the prevalence of ARI among the two groups were not statistically different due to the overlapping of the 95% CI of the prevalence. 

-The arguments are lack of scientific reasoning and reference?

Response: We tried to make the discussion more strong by adding several references. Thank you. 

- Implications were missed and what is the strength of your study?

Response: Yes, this is very key comment. Our study was a comparative cross-sectional, which was not as such a strong study design; however, implication of the study was further expanded as you can see in page 17 and 18. 

On the Conclusion

-The study revealed that the overall proportion of ARI among street sweepers and door-to-door waste collectors is high. what is your standard to say high/low? you should ope-rationalize it.

Response: Sure, it sounds the way our conclusion written is confusing. We updated the conclusion please see the updated version. See in page 18. Many thanks. 

-The difference in the proportion of ARI among street sweepers compared to door-to-door waste collectors was not statistically significant where do you get it? do you have data? but you did not present it in the result section and it is not your scope.

Response: We provided the result section within the result and please see the updated version of the manuscript. Yes, we have the data and it is our main findings since during comparative study, testing the difference or no difference of the prevalence is a must task to be done (See in page 13 from lines 269 to 277). 

-Being a street sweeper or a door-to-door waste collector was not significantly associated with ARI. where do you got it?

Response: We deleted that information and thank you for finding out such errors. 

- you are recommend that the municipality regularly provides these workers with adequate and quality PPEs, motivates and monitors the workers to use their PPEs while they are on duty an provides pre-employment safety training. are these significant finding affecting ARI? .

Response: It is our result section in Model I, Model II or pooled analysis of Model III. That is way we recommended as a solution. However, we adjust the conclusion based on the pooled analysis and the significant factors were not sue of mask and use of wood/coal and please see the revised version of the conclusion (See the updated version 8 and 9).

-you also recommended that these workers use a clean energy source like biogas and solar energy for cooking. what does it mean? are you considered biogas and solar energy as factors affecting ARI? In general, you should recommend according to your finding and practicality or applicability of result not from ground science.

Response: In our study, use of wood/coal was associated with ARI and that is why we recommended not to use wood/coal rather than use a clean energy source like biogas and solar energy for cooking, which is valid recommendation. We agree that our recommendation should be based on the findings. 

-you did not incorporate declarations in your manuscript as well.

Response: For PLoS ONE declaration is written online submission not within the manuscript. 

Reviewer #3

You raised an important issue which is a common problem of developing countries including Ethiopia. Overall, the document is well written but there are grammatical and writing errors which has to be corrected in addition to the following specific comments and questions.

Response: Thank you very much for duly acknowledging our work and we accepted all of your concerns and addressed them carefully herein. 

Title

• Your title is too long. Make it short. Avoid “A Comparative Cross-Sectional Study” from the title. It is not as such strong study design. It may not attract readers to go through the whole paper.

Response: Thank you and we deleted the design from the title to make it short. 

Abstract

• Remove abbreviations like ARI, CI and so on from abstract.

Response: We used the full name of acute respiratory infection than ARI in the abstract. However, since CI and AOR used several times, and not making the readers more boring with the text we used the abbreviations after once used both the full name and the abbreviation. CI and AOR are most commonly used the abbreviation form in the abstract. If we sued the full name of CI and AOR, the size of the abstract word will increase. 

• Methods:

The sampling technique is not mentioned.

Response: Thank you for this key comment. We briefly stated the sampling technique and please see the updated version and Figs, 1, 2 and 3 for further information. 

Instead of using the term bi-variate, use the term bi-variable. They are different concepts.

Response: We changed as suggested and please see the updated version. Thank you. 

• Results:

No need of writing 95% CI for associated factors in abstract section. Mentioning which variables were significant is enough.

Response: If we write in such a way as you suggested, it will become similar with the texts of the conclusion section. We prefer putting the main findings of the multivariable analysis is good for readers. However, we accept you comment and for clarity purpose, we prefer to put the 95 % CI for AOR and others. 

How the overall proportion of ARI could be the same (42.9%) with the proportion of ARI among street sweepers while there is difference in proportion of ARI among street sweepers (42.9%) and door to door waste collectors (36.9%)?

Response: Sorry for the error, we found that we were wrong in putting the exact prevalence rate. The overall prevalence means the average of the prevalence among the two groups. (See the abstract)

• Conclusion:

Don’t use at higher risk of ARI since your study design doesn’t allow you to measure risk.

Response: We appreciate your feedback and w deleted using risk throughout the paper. 

Background:

• Use terminologies uniformly like proportion and associated factors throughout the paper. Don’t mix up with prevalence and risk factors.

Response: Thank you, we revised to be consistent with rather than mixing up. 

Materials and Methods:

• Sample size and sampling procedure:

The assumption for allocating the sample size for each micro enterprise in both groups is not clear. Why you include all?

Response: Since the number of micro enterprise are small, about 10, then we considered all since the total source population 750 working in 10 enterprises. That is way we included all. 

• Data collection and measurements:

Change risk factors by associated factors

Response: We changed.

• Ethics approval and consent to participate:

How did you maintain confidentiality? specify it.

Response: We keep confidentially by did not recording names and other identifiers of the study participant (See the revised version of the ethics statement in page 8 and 9 from lines 199 to 206). 

Results:

Grammatical and write up errors should be corrected.

Response: we tried our best to avoid any grammatical and write up errors. 

• Characteristics of participants:

Instead of characteristics of participants, make it characteristics of street sweepers and door-to-door waste collectors.

Response: Thank you, we made it as suggested (See the result section)

• Proportion of ARI and ARI Symptoms:

“The proportion of ARI among street sweepers was 48.8 percent with 95% CI (37.3, 64.8) and among door-to-door waste collectors 36.9% with 95% CI (27.4, 46.4). The overall proportion of ARI among studied population was 42.9% with 95% CI (35.1, 50.0)”. This is different from what you write in the abstract section. Correct your writing error.

Response: Thank you for detecting such mistake and we did the revision accordingly for making the abstract in line with the body of the paper. 

• Factors Associated with ARI:

Factors for two groups are not clearly indicated. You should clearly mention which factors were significant among street sweepers and door to door waste collectors. That is expected from your study.

Response: Yes, this is very nice comment indeed. In the result section we found factors associated with ARI among street sweepers alone, among door-to-door waste collectors along and for both street sweepers and door-to-door collectors. Please see the result section in page --- and Tables 6, 7, and 8. 

Discussion:

• The comparison of the proportion of ARI made is not appropriate; children are compared with adults.

Response: Yes, we did this purposely due to the fact that ARI is most common in children than adults, where we faced lack of studies among adults. We tried to minimize such issues despite we are unable to make discussion due to lack of studies among adults regarding ARI. We noted this in the limitation section and please see the limitations of the updated version in page 13 to 14. 

Conclusions:

• You concluded that the overall proportion of ARI among street sweepers and door-to-door waste collectors is high. As you mentioned, the only comparable study done at Addis Ababa showed that the proportion of street sweepers with respiratory symptoms was 68.9%. So, how did you say that?

Response: Thank you for this key comment. We updated the conclusion by avoiding such confusion of high (See the conclusion in page 18 from lines 397 to 404). 

Table 1: Change inappropriate wording like illiterate

Response: We changed illiterate by cannot read and write. Thanking you. 

Table 2: Properly write the title by mentioning street sweepers and door-to-door waste collectors

Response: We mentioned street sweepers and door-to-door waste collectors

Reviewer #4

1. From the abstract do not use abbreviations. 

Response: Thanks, we minimized the use of abbreviation in the abstract, but we used CI and AOR since they are more frequently used in the abbreviation to minimize size of the abstract. 

2. Sample size is too small in amount, how did you conclude?

Response: Dear reviewer, we agree that our sample size looks small. However, as you can see in the sample size determination procedures using double population formula, the sample size was determined based on scientific assumptions and we hope that it is not a major issues. Nevertheless, we agree that higher sample size always acceptable. 

3. Your discussion is superficial

Response: Thank you for this key comment and we tried to strengthen the discussion and please kindly visit the revised version. The problem with our discussion was there was lack of study among adults and ARI study was mainly common among children. This limited as unable to discuss as we want. Even some of our discussion was made based using ARI studies among children, which is not valid but still good to see since only few studies available among adult populations. I hope that you understood the situation. Many thanks. 

4. Minimize Page Numbers

Response: We tried to minimize the page numbers despite the comparative study nature always taking more ideas compared to the cross-sectional survey. We feel that it is the ideas that contain matters than the number of pages. We tried our best to address your concerns. 

5. it has grammatical error, please check it

Response: Thank you for detecting such errors and we addressed these concerns. Please see the revised version. 

 We would like to thank the reviewers and editors for evaluating our manuscript. We have tried to address all the concerns in a proper way and believe that our paper has been improved considerably. We would be happy to make further corrections if necessary and look forward to hearing from you all soon. 

I hope that the revised manuscript is accepted for publication in PLoS ONE. 

Sincerely yours,

Metadel Adane (PhD in Water and Public Health)

---

## [Decision Letter · Decision Letter 1]

30 Apr 2021

Prevalence and Associated Factors of Acute Respiratory Infection among Street Sweepers and Door-to-Door Waste Collectors in Dessie City, Ethiopia

PONE-D-20-05977R1

Dear Dr. Adane,

We’re pleased to inform you that your manuscript has been judged scientifically suitable for publication and will be formally accepted for publication once it meets all outstanding technical requirements.

Kind regards,

Miguel A. Fernández, Ph.D.

Academic Editor

PLOS ONE

Additional Editor Comments (optional):

Reviewers' comments:

Reviewer's Responses to Questions

**Comments to the Author**

1. If the authors have adequately addressed your comments raised in a previous round of review and you feel that this manuscript is now acceptable for publication, you may indicate that here to bypass the “Comments to the Author” section, enter your conflict of interest statement in the “Confidential to Editor” section, and submit your "Accept" recommendation.

Reviewer #1: All comments have been addressed

Reviewer #2: All comments have been addressed

Reviewer #3: All comments have been addressed

2. Is the manuscript technically sound, and do the data support the conclusions?

Reviewer #1: Yes

Reviewer #2: Yes

Reviewer #3: Yes

3. Has the statistical analysis been performed appropriately and rigorously? 

Reviewer #1: Yes

Reviewer #2: Yes

Reviewer #3: (No Response)

4. Have the authors made all data underlying the findings in their manuscript fully available?

Reviewer #1: Yes

Reviewer #2: Yes

Reviewer #3: (No Response)

5. Is the manuscript presented in an intelligible fashion and written in standard English?

Reviewer #1: Yes

Reviewer #2: Yes

Reviewer #3: Yes

6. Review Comments to the Author

Reviewer #1: (No Response)

Reviewer #2: Authors have processed my feedback appropriately, it increased the quality of the article significant. Good luck with your research!

Reviewer #3: The paper has been improved very well. All the comments are properly addressed.

• The title is shortened as suggested

• The abstract is well corrected

• The sampling technique and procedures are briefly stated

• The results are written properly. Percentage errors are well corrected

• Terms such as prevalence and risk factors are avoided, and proportion and associated factors are uniformly used throughout the document

• The assumption for allocating the sample size for each micro enterprise is clearly justified

• The issue of confidentiality is well addressed

• Grammatical and write up errors have been improved

• Factors for street sweepers and door to door waste collectors as well as factors for the overall participants are clearly indicated in separate tables

• The comparison of the proportion of ARI among children and adults are justified

• The conclusion is modified

• The table titles are corrected

By now, I will be happy if the paper is published at PLOS ONE.

7. PLOS authors have the option to publish the peer review history of their article (what does this mean?). If published, this will include your full peer review and any attached files.

Reviewer #1: No

Reviewer #2: **Yes: **Erkihun Tadesse

Reviewer #3: **Yes: **Getaw Walle Bazie

---

## [Editor Report · Acceptance letter]

6 May 2021

PONE-D-20-05977R1 

Prevalence and Associated Factors of Acute Respiratory Infection among Street Sweepers and Door-to-Door Waste Collectors in Dessie City, Ethiopia: A Comparative Cross-sectional Study 

Dear Dr. Adane:

I'm pleased to inform you that your manuscript has been deemed suitable for publication in PLOS ONE. Congratulations! Your manuscript is now with our production department. 

Kind regards, 

on behalf of

Dr Miguel A. Fernández 

Academic Editor

PLOS ONE